# Cell Immunotherapy against Melanoma: Clinical Trials Review

**DOI:** 10.3390/ijms24032413

**Published:** 2023-01-26

**Authors:** Ivan Y. Filin, Yuri P. Mayasin, Chulpan B. Kharisova, Anna V. Gorodilova, Kristina V. Kitaeva, Daria S. Chulpanova, Valeriya V. Solovyeva, Albert A. Rizvanov

**Affiliations:** Institute of Fundamental Medicine and Biology, Kazan Federal University, 420008 Kazan, Russia

**Keywords:** immunotherapy, melanoma, clinical trials, T-cells, B cells, NK cells, dendritic cells, macrophages

## Abstract

Melanoma is one of the most aggressive and therapy-resistant types of cancer, the incidence rate of which grows every year. However, conventional methods of chemo- and radiotherapy do not allow for completely removing neoplasm, resulting in local, regional, and distant relapses. In this case, adjuvant therapy can be used to reduce the risk of recurrence. One of the types of maintenance cancer therapy is cell-based immunotherapy, in which immune cells, such as T-cells, NKT-cells, B cells, NK cells, macrophages, and dendritic cells are used to recognize and mobilize the immune system to kill cancer cells. These cells can be isolated from the patient’s peripheral blood or biopsy material and genetically modified, cultured ex vivo, following infusion back into the patient for powerful induction of an anti-tumor immune response. In this review, the advantages and problems of the most relevant methods of cell-based therapy and ongoing clinical trials of adjuvant therapy of melanoma are discussed.

## 1. Introduction

Melanoma is one of the most dangerous and aggressive types of cancer. The primary risk factor for melanoma is exposure to UV rays, which leads to the direct DNA damage and malignant transformation of melanocytes located in the basal layer of the epidermis [1]. In response to UV exposure, keratinocytes, the main cells of the epidermis layer of the skin, produce α-melanocyte-stimulating hormone, which binds to the melanocortin 1 receptor (MC1R) on melanocytes, stimulating melanin synthesis, which acts as a protector against UV radiation [2,3].

People with a low level of MC1R activity accumulate more mutations due to the increased damaging effects of UV rays. Skin cancer can occur when mutations are accumulated in the sensitive regions of genome [2]. According to the National Institute of Oncology, melanoma was 5th most common type of cancer in 2022 and the incidence rate of melanoma is increasing every year [4].

Such treatment approaches as excisional surgery, chemotherapy, radiation therapy, immunotherapy with checkpoint inhibitors, and targeted therapy (signal transduction and angiogenesis inhibitors, oncolytic viral therapy) are commonly used for melanoma therapy [4]. However, a large number of clinical trials of new methods of melanoma treatment are being conducted today. In this review, cell-based therapy of melanoma and its benefits are discussed and compared to the results of ongoing clinical trials for the new treatment approaches.

## 2. Melanoma

Melanoma is a skin cancer caused by the malignant transformation of melanocytes. The first case of this disease was documented in the Western medical literature in the 17th century, namely in the works of Highmore and Bonet, who described melanoma as “deadly black tumors with metastases and black fluid in the body” [5]. Today, melanoma is considered the most dangerous and aggressive type of skin cancer, accounting for about 90% of deaths among skin cancers [6]. Risk factors that are responsible malignant transformation of melanocytes include age, gender, race, genetic predisposition, environmental conditions, and the number of moles (nevi) [7]. Nevi are benign accumulations of melanocytes, a number of which can be hereditary or acquired. There are data that indicate a positive correlation between the development of melanoma and nevus presence and number [8].

According to one of the existing classifications, melanoma is divided into non-cutaneous and cutaneous melanoma, the latter of which is the most common (about 92%) and often arises due to genetic predisposition or DNA damage resulting from UV exposure. For example, such molecular pathways as MAPK and PI3K/AKT are commonly dysregulated in melanoma. Moreover, mutations in cyclin-dependent kinase inhibitor 2A (CDKN2A) are common among human cancers including melanoma [9]. Hereditary factors also play a significant role in the development of melanoma. Pigmentation characteristics (e.g., xeroderma pigmentosum, oculocutaneous albinism), response to ultraviolet light, and nevus number contribute to a patient’s overall melanoma risk [10]. Cutaneous melanoma can be caused as a result of chronic sun damage and non-chronic sun damage, respectively [9]. Non-cutaneous melanoma typically occurs in areas with low UV exposure, such as the choroid and iris, the ciliary body (uveal melanoma), mucosal tissue, and acral tissue. This type of melanoma is much less common than cutaneous melanoma (about 10%). Acral melanoma can be found in the areas of skin without hairline (e.g., palms, soles, space under the nails) and is most common in older people with dark skin color [11]. Mucosal melanoma is the rarest type of melanoma and is associated with melanocytes located in the mucous membranes of the gastrointestinal, genitourinary, and respiratory tracts [12].

There are several treatment approaches for melanoma. In addition to surgical excision, chemotherapy, and radiotherapy, various types of immunotherapy are of particular importance (Figure 1), especially in cases of metastatic melanoma, since the five-year survival rate of such patients is only about 25% [12].

## 3. T-Cell Therapy

Adoptive cell therapy (ACT) is a type of immunotherapy for cancer that uses genetic engineering to improve the ability of T-cells to recognize and destroy tumor cells. There are three main types of T-cell therapy, depending on different mechanisms of action: tumor infiltrating lymphocyte (TIL) therapy, chimeric antigen receptor (CAR) T-cell therapy, T-cell receptor (TCR)-based therapy, and natural killer T-cell (NKT) therapy. Another newly defined T-cell subset is stem cell-like memory T-cells (T_SCM_). T_SCM_ are self-renewal and log life span cells with pluripotency potential and rapid differentiation into effector T-cells. T_SCM_ is developed by different methods through several signal pathways from naive T cells [13]. It is worth noting that the definite T-cell phenotype may make an impact in mediating anti-cancer response. Krishna et al. underscore the importance of T-cell phenotypes in ACT response and identified a memory-progenitor stem-like TIL associated with complete cancer regression, in particularly against melanoma. They suggest that T_SCM_ might provide opportunities for the development of more effective T cell-based immunotherapies [14]. There are several studies including CAR-T-modified T_SCM_ that showed that T_SCM_ have good clinical application prospects in antitumor immunotherapy. However, in spite of good preclinical evidence in antitumor responses, these T-cells have still inefficient activity against solid tumors. Additionally, the clinical application of T_SCM_ cells is hindered because they are relatively rare in the circulation [15]. However, T_SCM_ are a promising method in human adoptive immunotherapy that, at this early date, should be a priority research area.

### 3.1. TIL Therapy

TIL therapy is type of immunotherapy in which TILs are isolated from the tumor site by biopsy or surgery [16]. These cells are cultured in vitro with interleukin-2 (IL-2), which increases the antitumor response due to induction of CD4^+^ T-cell proliferation and differentiation into T-helpers [17]. It has also been shown that IL-2 can increase the antitumor activity of CD8^+^ T-cells and cytotoxic function of NK cells [18]. The prepared vaccine is infused back into the patient [16]. It takes 5–7 weeks to produce a TIL-based drug, which requires specialized equipment and trained staff [19].

TILs are a polyclonal population, which consists of CD20^+^ B cells, CD3^+^ T-cells (CD4^+^ and CD8^+^ T-cells), and FOXP3^+^ regulatory T-cells [20], whose prognostic role in melanoma is currently under investigation [21].

Steven Rosenberg was the first person who demonstrated the effectiveness of a TIL-based method in the treatment of melanoma in 1988 [22]. Twenty participants with metastatic melanoma who took part in Steven Rosenberg’s clinical trial were infused with a cultured suspension of TILs and IL-2. As a result, tumor regression was observed in 9 out of 15 patients who had not previously been received IL-2 and in 2 of 5 patients who were previously unsuccessfully treated with IL-2. Tumor regression was observed in the lungs, liver, bones, skin, and subcutaneous adipose tissue. It was noted that TIL efficiency largely depends on prior treatment with cyclophosphamide or radiation therapy and simultaneous administration of IL-2. It is important to note that most TILs had a CD3^+^ phenotype, but the numbers of CD4^+^ and CD8^+^ cells varied between patients [22].

The FDA has assigned the TIL-based drug ITIL-168 as an orphan drug for the treatment of patients with stage IID to IV melanoma in April 2021. The results of the treatment of 21 patients with this drug in combination with cyclophosphamide and fludarabine were presented at the American Association for Cancer Research meeting in 2021 [23]. Complete response (CR) and partial response (PR) were noted in 19% and 48% of patients, respectively. In addition, stable disease (SD) was observed in 19% of patients, and 14% had an increase in tumor size and progressive disease (PD). Patients who responded had a median time to response of 1.7 months. With the median follow-up 52.2 months, responses were durable for 23% of the 14 responding patients, lasting more than 30 months after TIL infusion [23]. Nowadays, ITIL-168 is currently in a phase II clinical trial, with the primary completion date March 2024 [NCT05050006].

Obsidian Therapeutics obtained permission from the FDA to begin a phase I clinical trial of OBX-115 [NCT05470283] in July 2022. This drug is TILs expressing membrane-bound IL-15, which is necessary for increasing the efficiency and duration of the immune response. In addition, therapy with this drug does not need additional administration of IL-2, which is required for early TIL-based drugs [24].

Another FDA submitted drug is lifileucel (LN-144) in patients with advanced melanoma [NCT02360579] [25]. In the phase 2 clinical trial, patients with unresectable or metastatic melanoma that was stage IIIC or IV who had confirmed radiologic progression and who had been previously treated with checkpoint inhibitor(s) and BRAF ± MEK targeted agents received lifileucel with IL-2. The objective response rate (ORR) in pooled patient cohorts 2 and 4 (*n* = 153) was of 31% (95% CI, 24.1%–39.4%) by independent review committee (IRC) assessment. CR and PR were observed in 5% and 26% of patients, among those who responded to the treatment. Median overall survival and progression-free survival were 13.9 and 4.1 months, respectively [26].

One of the main advantages of TIL-based therapy is its ability to treat cancer patients in the terminal stage who have been received different types of antitumor therapy [27]. However, TIL-based therapy has side effects that are associated with combined administration of IL-2 and chemotherapy [27]. Third-level side effects were described in some clinical trials. Non-myeloablative lymphodepleting chemotherapy can lead to some abnormalities of blood cellular components, such as transient cytopenia (neutropenia, lymphopenia, and prolonged decrease in CD4^+^ T-cells) [28]. In some of these cases, granulocyte colony-stimulating factor and blood transfusion were used to improve the patient’s condition. Non-hematological side effects of TIL therapy include diarrhea, hyperbilirubinemia, and neurotoxicity, which are associated with the fludarabine administration [28]. Allergy reactions after TIL injection, which may manifest as cytokine release syndrome (CRS), accompanied by fever and skin reactions, are also reported [29]. However, the toxic effect of IL-2 was reduced in patients treated with lymphodepleting chemotherapy, in which lymphocytes were the main source of cytokine production [30]. Supportive therapy must be performed for the prevention and treatment of side effects.

Nowadays, more than 100 clinical trials of TIL-based therapies of melanoma are under investigation, seven of which have been completed and have reported results: NCT02354690, NCT02379195, NCT00314106, NCT00509288, NCT01118091, NCT00513604, NCT00096382.

### 3.2. TCR Therapy

TCR therapy is an injection of genetically modified T-lymphocytes that express TCRs specific to certain tumor antigens. TCR is heterodimer consisting of α- and β-chains, which are covalently linked through disulfide bonds facilitated by conserved cysteine residues located in each chain [31]. TCR α/β heterodimers further bind with CD3ϵ/γ/δ/ζ subunits to form a functional receptor [32]. TCRs recognize enzymatically cleaved peptides that are presented at the cell surface by major histocompatibility complex (MHC) molecules. TCR binding to corresponding MHC leads to the phosphorylation of immunoreceptor tyrosine-based activation motif (ITAM) in intracellular CD3 subunits [33]. This leads to T-cell proliferation, cytokine secretion, and cytolysis, due to secretion of perforin and granzyme [34]. In TCR therapy, T-cells are genetically modified to express the α- and β-chains of the TCR, which confer the required specificity.

Manufacturing genetically engineered T-cells includes several stages. Firstly, T-cells are isolated from the patient’s blood or tumor tissue [35]. Then, after their activation and amplification in vitro*,* T-cells are modified by transduction with lentiviral or retroviral vectors for specific TCR expression [36]. After amplification and quality control, TCR-T-cells are injected into the patient [35].

Targets of TCR-T-cell therapy include glycoprotein gp100 (PMEL, melanoma), carcinoembryonic antigen (CEA, colorectal cancer), melanoma antigen recognized by T-cells 1 (MART-1, melanoma), melanoma-associated antigen 3 (MAGE-A3) (melanoma/multiple myeloma), and New York esophageal squamous cell carcinoma 1 (NY-ESO-1) (melanoma/synovial cell sarcoma) [37].

The main advantage of this therapy is that intracellular proteins can also be presented by MHC molecules, therefore TCR-T-cells can target any tumor-specific or tumor-associated intracellular protein [35]. However, neurological toxicity [38], and non-targeting toxicity following gene therapy in which there is an antigen-specific attack on non-tumor-specific tissue that expresses the tumor-associated target antigen, have been observed in some studies [39,40]. In addition, hypercytokinemia resulting from the activation of the immune system is typical for both TCR therapy and T-cell-based therapy [41,42]. The above mentioned effects are clinically manifested as systemic inflammatory response syndrome, characterized by fever, tachycardia, hypotension, vasodilation, and capillary leak [39]. CRS can also be developed after T-cell infusion during TIL-based therapy. Severe forms of CRS can lead to shock and multiple organ failure [28].

Tebentafusp **(**KIMMTRAK) became the first TCR-based drug for juvenile melanoma approved by FDA in January 2022. The MHC target for this drug is A*02:01, which presents melanoma antigen gp100 to T-cells [43]. Nowadays, more than 70 clinical trials of melanoma therapy using T-cells with genetically modified TCR are registered. Some of them are completed and have results: NCT00910650, NCT00509496, NCT00509288, NCT01350401, NCT00610311, NCT00612222, NCT02062359.

### 3.3. CAR T-Cell Therapy

CAR T-cell therapy is based on the use of the T-cell chimeric antigen receptor. CARs are engineered synthetic receptors that redirect lymphocytes to recognize and eliminate cells expressing specific antigens [44]. CAR binds with the target antigens, expressed on the cell surface, independently from the MHC receptor, which leads to the enhanced activation of T-cells and strong anti-tumor immune response [45]. There are five generations of CAR T-cell therapy depending on the amount and combination of co-stimulating molecules [46]. In comparison with TCR-therapy, CAR-based therapy has a less broad spectrum of protein antigens [47]. Preclinical trials have identified such antigens as CD16, CD126, CD70, B7-H3 (CD276), human epidermal growth factor receptor 2 (HER2), vascular endothelial growth factor receptor 2 (VEGFR-2), gp100/human leukocyte antigen A2 (HLA-A2) complex, chondroitin sulfate proteoglycan 4 (CSPG4), disialoganglioside GD2, and GD3, which can be used for CAR-T therapy [48].

The CAR T-cell manufacturing process is divided into five stages. The first step is T-cell isolation from the cancer patient. The second step is T-cell modification to express CAR, which can recognize tumor cells and activate T-cells. In the third stage, CAR T-cells are cultured ex vivo and stimulated by cytokines to produce more CAR T-cells. The fourth step is injection of cultivated CAR T-cells into the patient at the appropriate dose [49]. The entire process of manufacturing takes about 3 weeks, with average of 2 weeks to prepare CAR T-cells [48,49].

CAR T-cell therapy is associated with unique Grade 1–2 side effects, the most common of which is the development of hypercytokinemia [50]. The affinity of CAR T-cells causes rapid release of a large number of cytokines, such as TNF-*α*, IL-1, IL-2, IL-6, interferon (IFN)-*α*, and IFN-*γ*, which can cause acute respiratory distress syndrome and multiple organ failure [46]. Most cases of CRS are manifested as a flu-like disorder with fever, malaise, headache, tachycardia, and myalgias [28]. The severity of CRS correlates with the tumor burden [51]. Additionally, neurological disorders may develop during the therapy, the pathogenesis of which is not completely explained [35]. However, antigen receptors on the T-cell surface after genetic modification are aimed at identifying specific antigens on tumor cells’ surface [35]. However, these antigens also could be expressed in normal tissues, leading to normal cell damage [52]. In 2018, recommendations for diagnostic, grading, and treatment of toxicities associated with CAR T-cell therapy, CARTOX (CAR T-cell therapy-associated toxicity), were published [50]. These guidelines also include a list of lethal events observed to date in CAR T-cell trials.

Nowadays, only 12 clinical trials dedicated to CAR T-cell-based therapy of melanoma have been conducted: NCT04119024, NCT03893019, NCT05190185, NCT05117138, NCT02107963, NCT03638206, NCT04483778, NCT04897321, NCT02830724, NCT03635632. The results were published only for the one of them [NCT01218867]. The study involved 24 patients with metastatic melanoma. They received anti-VEGFR2 gene modified white blood cells. Patients received CAR T-cells in different quantities with a low or high concentration of IL-2. Research was terminated due to a lack of objective response. In total, 23 of 24 patients had progressive disease, and 23 of 24 patients had serious side effects caused by the therapy, with 5 of 24 being in critical condition [53].

### 3.4. NKT-Cell Therapy

The NKT-cell population has many features similar to conventional T-cells, such as development in the thymus and expression of TCR. However, TCRs of NKT-cells recognize lipid antigens presented by CD1d and induce a rapid immune response consisting of the production of large amounts of cytokines and chemokines [38]. There are two types of NKT-cells: ‘invariant’ cells (iNKT) containing an invariant TCRα chain connected with a heterogenous TCRβ chain; and ‘variant’ NKT cells which lack an invariant chain [38]. The ligand for iNKT-cells is α-galactosylceramide (α-GalCer), a sphingolipid that was, for the first, time isolated from the marine sponge *Agelas mauritianas* in 1994 [54]. α-GalCer is widely used because of its ability to induce significant activation of mice and human iNKT-cells [55].

Morita et al. demonstrated a high anti-tumor response against metastatic B16 melanoma cells in mice injected with free α-GalCer [56]. In addition, α-GalCer had synergetic anti-tumor effects when co-administrated with another chemotherapy agent, Adriamycin [54].

There are only two clinical trials of melanoma therapy using NKT-cells, NCT02619058 and NCT02619058, which have no published results.

Exley et al. began a phase I clinical trial of iNKT-cells for melanoma in 2017 [57]. Nine patients received infusions of iNKT-cells that were isolated after leukapheresis. The median follow-up of this patient cohort was 63 months (from 53 to 85 months). As a result of the therapy, three patients have died, three showed no progression of the disease at 53, 60, and 65 months, and three received further treatment and were alive at 61, 81, and 85 months. Toxicity associated with the therapy was limited to Grades 1 and 2. Subfebrile fever after infusion, local skin reaction, and constitutional symptoms, which include weight loss, fever, headache, fatigue, dyspnea, and malaise were reported as the adverse events [57].

CAR can also be used for iNKT-cells since they mediate a protective immune response against malignant neoplasms through different mechanisms [58].

## 4. B Cell Therapy

B cell therapy is a promising approach for the treatment of melanoma. Unlike CAR-T-cell therapy, immunotherapy based on B cells is still poorly investigated. Role of B cells in the tumor microenvironment has a dual character [59]. On the one side, B cells provide an antitumor immune response through secreting tumor-specific antibodies and therefore suppressing tumor growth. On the other side, protumor properties are related with the function of a specific B cell population, such as regulatory B cells (Bregs) [60]. Bregs release cytokines, which inhibit functions of cytotoxic T-cells and suppress immune response.

Different variations of immunotherapy based on B cells are relied on for their antitumor functions. Because of ability of B cells to synthesize antibodies, the anti-cancer response is provided by a variety of mechanisms: phagocytosis by macrophages, activation of a complement system, inactivation of NK cells, and activation of dendritic cells (DCs) [61]. A number of drugs such as ipilimumab, nivolumab, and pembrolizumab are monoclonal antibodies approved by the FDA for the treatment of metastatic melanoma. These drugs show a significant number of responses and low recurrence rate [62].

The spectrum of action of B cells in the tumor microenvironment is more multifaceted than only synthesis of antibodies. B cells can also be seen in the role of antigen-presenting cells that express MHCII on the cell surface, taking part in CD8^+^ and CD4^+^ T-cell activation [63]. B cell receptor (BCR)-mediated antigen internalization occurs primarily by clathrin-mediated endocytosis [64]. After that, endocytic BCR:antigen complexes are crossed with newly synthesized MHC on the way to cell surface, then, an antigen is presented by MHC on the cell membrane [65,66]. It is worth noting that the entire process of antigen uptake is more effective with the CD40 receptor. Cells cultured with the CD40 ligand and IL-4 were named CD40B and are of particular interest as antigen-presenting cells, which have several advantages over DCs [63].

B cells secrete different cytokines and chemokines, which modulate immune reactions. B cell therapy has great potential because of low toxicity. B cells are primarily of interest from the point of view of the production of antibodies, which are in clinical trials or have already approved by FDA. Despite the efficiency of these antibody-based drugs, there are several pre- or clinical trials, in which B cells were involved as anticancer cell drugs for melanoma treatment. In one preclinical study in mice C57BL7/6, CD40B cells were modified by the electroporation of multiple mRNAs encoding immune stimulatory molecules, but treatment did not slow the growth of B16.F10 melanoma [67]. In another preclinical study, mice were vaccinated with CD40B cells with the TRP2 pulse antigen and tumor growth was significantly reduced as early as day 11 compared to negative controls [68].

A hybrid cell vaccine was used in a clinical trial [69]. The vaccine was generated by fusion of the autologous tumor cells with activated allogenic B cells, which were isolated from a donor’s peripheral blood. The resulting hybrid cells express MHCI molecules. Sixteen patients with advanced stage metastatic melanoma participated in the study. Of these, 6% of the patients had CR, PR was seen in 6%, and SD in 31%. In total, 88% of patients had NAE (number of adverse events) classified as Grade I and II of toxicity.

The dual nature of B cells makes the use of this type of therapy in clinical practice complicated. Antibodies can form circulating immune complexes (CICs) which bind to the Fcγ receptor on immunosuppressive myeloid cells and can contribute to vascularization of the tumor. In addition, it is worth paying attention to different populations of B cells and their influence on tumor formation. There is also the problem related to autoantigens that can express on both tumor cells and normal cells. Consequently, targeting autoantigens can lead to the development of toxic side effects [59].

## 5. NK Cell Therapy

NK cells are the effector cells of the innate immune system and NK cell-based immunotherapy is another promising alternative to CAR-T therapy for the treatment of melanoma. The advantages of this ACT include low infusion toxicity, insensitivity to tumor escape via the MHC pathway, as well as a variety of methods for obtaining NK cells, which will be discussed below. All these advantages of NK cells make it possible to rest great hopes on NK therapy. However, the immunosuppressive properties of the tumor can delimitate NK therapy and various populations of immune system cells, as well as exosomes secreted by melanoma cells, can exert an inhibitory effect in the tumor microenvironment [70,71]. Additionally, at the moment, it has been shown that, in solid tumors, NK cells work worse than in various leukemias, since there is an additional problem associated with the insufficient infiltration of NK cells into the tumor.

NK cells use several mechanisms to kill target cells, and, unlike B and T-cells, their receptor formation does not involve VDJ recombination, and NK cell activation depends on complex receptor:ligand interactions [72]. The main mechanism of action of NK cells is the destruction of target cells by lytic granules (perforin and lysine), which is carried out through the interaction of several receptors:ligands, primarily killer immunoglobulin-like receptors with MHCI. If the target cell does not express the MHC molecule, this is a signal for the activation of NK cells. Another way is death receptor-mediated apoptosis, which is initiated by activation of caspases 8 and 10 induced by the binding of death receptors to ligands. NK cells, via CD16, are able to bind to the Fc region and destroy malignant cells opsonized by immunoglobulins through antibody-dependent cell-mediated cytotoxicity [73,74].

Considering NK cells as a potential immunotherapeutic agent, scientists have investigated various methods of NK cell isolation from various sources, such as cord blood, peripheral blood mononuclear cells, the NK92 cell line, and differentiated induced pluripotent stem cells (iPSCs) [75].

Various clinical trials are based on different types of NK therapy. In addition to classical autologous and allogeneic NK therapies, they are focused on developing new ways to increase the effectiveness of vaccines and create new options for immunotherapy, such as CAR-NK and TCR-NK. Parkhurst et al. used autologous NK cells to treat patients with melanoma in 2011 [76]. In the study, seven patients received NK cell infusions after chemotherapy. No clinical effects were observed, which, according to the authors, is due to the low expression of NKG2D on the persistent NK cells, which is necessary for the implementation of the cytotoxic function.

Arai et al. conducted the first stage of a clinical trial using the NK92 cell line in 2008 [77]. The study involved one person with melanoma, who had a slight improvement after the infusion, but soon died due to the progressive disease.

Trial NCT03841110 combined NK cells differentiated from iPSCs and immune checkpoint inhibitors (atezolizumab or nivolumab). The authors currently report ORR in 24% of patients and all causes mortality (ACM) in less than 3% of patients.

Current clinical trials are focused on various NK cell combination vaccines. In the NCT00720785 study, autologous NK cells were used in combination with bortezomib, a proteasome inhibitor, which has been shown to increase NK cell-mediated cytotoxicity [78]. The study NCT05588453, which uses universal donor NK cells with TGFβ-imprinting (TGFbetai) in combination with temozolomide as a therapy, started in 2022. This therapy is aimed at shrinking tumors in patients with stage IV melanoma that has metastasized to the brain. At the moment, these trials are either not completed or the results have not been published, so we can only expect favorable results in the near future.

Although NK cells are promising for the treatment of cancer patients, there are a number of disadvantages associated with this type of adjuvant therapy, such as difficulties in ex vivo clinical expansion, limited in vivo persistence, insufficient infiltration of solid tumors, and immunosuppressive mechanisms of the tumor microenvironment [79].

## 6. Dendritic Cell Therapy

This type of cells named "dendritic" were, for the first time, described by R. Steinman in 1973 [80]. Almost two decades later, in the first half of the 1990s, there was a significant rise in scientific interest in immunotherapy based on DCs. This is because enough data had accumulated to confirm the antigen-presenting properties of DCs by the end of the 1980s [81]. DCs are professional antigen-presenting cells due to their ability to capture and process various antigens for their further presentation to effector cells [82]. In turn, effector cells (e.g., T-cells) have a direct or indirect suppressive effect on the pathogen or tumor cells [83]. Due to this ability, it is possible to use DCs for cancer immunotherapy [84] when tumor lysate [85], mRNA [86], inactivated tumor cells [87], synthetic peptides [88], and extracellular vesicles (EVs) [89] can be used as an antigen. This type of immunotherapy was widely developed by the early 2000s, and, at the moment, many clinical trials are ongoing or have already been completed.

### 6.1. Autologous DC-Based Therapy

Kyte et al. showed that during phase I/II clinical trials [NCT01278940] of a vaccine from autologous DCs loaded with autologous tumor mRNA against stage IV melanoma, the survival of patients with a subsequent immune response to the vaccine was increased more than expected. According to the results, multiple vaccinations over a short period of time (13 vaccines in 10 months) showed no short-term toxic effects. Longer follow-up (more than 11 years) also showed no long-term toxic effect. Based on the criteria presented by the authors, 52% of the participants (16 people) were immunosusceptible to the antigen immediately after vaccination. Speaking about the long-term antitumor response, it should be noted that SD was observed in 10% of patients (3 people), PR only in 3% of patients (1 person), and no evidence of disease (NED) in another 6% (2 patients). The remaining 81% of participants showed signs of PD [86].

In the study of Butterfield et al., a clinical trial [NCT01622933] evaluated the ability of an antigen-engineered DC vaccine to induce a polyclonal CD8^+^ and CD4^+^ T-cell response against three common melanoma antigens. The 35 vaccine recipients were randomly assigned to two groups, with or without high doses of IFN-α. The study showed that the addition of IFN-α failed to improve the immune or clinical response compared to the control group. Among all patients with measurable disease, PR was observed in 8% of patients (2 persons), SD in 33% (8 persons), while signs of PD were found in the remaining 59% of participants (14 persons). Only 36% (4 people) out of 11 surgically treated patients remained NED with a mean follow-up of 3 years. In the majority of vaccinated patients, an increase in the specific response of CD8^+^ and CD4^+^ T-cells to the vaccine antigen was observed. In 51% of patients, NAE Grade I–IV toxicity was observed, but only in 25.5% was this manifestation associated with the introduction of DC vaccines and did not go beyond Grade I toxicity [90].

In a phase I/II clinical trial of a DC vaccine in patients with metastatic cutaneous melanoma [NCT00053391], persistent vaccine-specific immune responses were obtained. DCs were obtained from monocytes and cultured in the presence of TNF-α, IL-1β, IL-6, and prostaglandin E2, and then loaded with HLA-A1- and HLA-A2-restricted tumor peptides, after which they were being injected intradermally at high doses for 2 years. As a result, IFN-γ-producing CD4^+^ and cytotoxic polyfunctional CD8^+^ T-cells were induced de novo or enhanced in most patients. At the time of publication of the research results, the incidence of OS in patients with unresectable metastatic melanoma was 19%, which is consistent with the survival observed in patients treated with ipilimumab. However, a similar result was achieved without any serious toxicity (no more than Grade II). Survival was significantly correlated with the development of intense reactions at the site of vaccine injection and with blood eosinophilia after the first series of vaccinations, indicating that the prolonged survival was a result of DC vaccination [91].

A large number of other clinical trials are still ongoing. Thus, a phase III study which will evaluate an active immunization during adjuvant therapy of patients with stage IIIB and IIIC melanoma using natural DCs pulsed with synthetic peptides (NCT02993315) is scheduled to be completed in 2024. The aim of this study is to determine whether natural DC vaccination improves NED compared to placebo-matched treatment.

In a number of clinical trials, the safety of DC vaccines [NCT04335890, NCT01946373, NCT01082198], the formation of an immune response [NCT01082198, NCT01331915, NCT00085397], an increase in recurrence-free survival [NCT02993315, NCT02301611], and moderate dose-limiting toxicity [NCT04335890, NCT01082198] were noted as an intermediate or final result.

### 6.2. Therapy Based on DC-Derived Exosomes

Despite the use of whole autologous DCs, EVs from DCs are also used for immunotherapy. This is a heterogeneous fraction of membrane-surrounded compartments constitutively secreted by dendritic cells and containing soluble proteins, small molecules, and nucleic acids [92]. The membrane of EVs has the same profile and expression of surface markers as the cell membrane of parental cells, due to which, the MHC-peptide complexes and co-stimulatory molecules on the surface of EVs are able to interact with target cells [93].

One feature of DC-derived EVs is their ability to activate effector cells of innate and adaptive immunity. This is due to the fact that DCs carry MHCI/II molecules, various costimulatory molecules, and other ligands on their surface, which provide interaction with T-cells and stimulate NK cell activation, causing a combination of factors to kill the tumor and counteract its immunosuppressive microenvironment. Furthermore, it has been shown that EVs derived from preliminary tumor antigen-loaded DCs can be internalized by other DCs and thereby induce antigen cross-presentation [94]. It has also been shown that a special group of EVs derived from DCs—DC-derived exosomes (Dex)—can be used to more accurately target the delivery of a processed and presented antigen to secondary lymphoid organs, in contrast to whole autologous DCs, which are predisposed to chemokine-dependent migration [93].

In the phase I clinical trial by Viaud et al., Dex were shown to have no effect on T-cell activation but were able to enhance NK cell recycling and cytotoxicity by interacting with the NKG2D ligand on the surface of Dex. As a result, the cytotoxic activity of NK cells was upregulated, as well as the recognition and destruction of tumor cells. Fifteen patients received four doses of the Dex-containing vaccine at weekly intervals, followed by an assessment of the functional activity of T- and NK cells within 7 weeks of the first Dex vaccination. While the pool of lymphocyte populations remained stable throughout the therapy, the proportion and absolute number of circulating NK cells was increased significantly after four Dex vaccines. The study of the NK cell phenotype in these patients with advanced melanoma showed that NKG2D levels were significantly reduced compared to healthy individuals, but significantly increased after injections of Dex vaccines. An evaluation of the cytotoxic activity of NK blood cells from patients was carried out to assess the functional significance of this observation. ORR was observed in 50% of patients. This is due to the increased cytotoxic activity of NK cells, which correlates with the level of NKG2D expression being increased to normal levels after the introduction of the Dex vaccine. However, in the other 50% of non-responders (NKG2D levels were still low), there was no increase in cytotoxic activity against the target cells. In 13% of patients (two persons) with confirmed tumor regression who continued treatment (Dex administration every three weeks for 6–10 months), the effector functions of NK cells remained elevated at later time points [95].

## 7. Macrophage-Based Cell Therapy

Macrophages are a heterogeneous population of innate immunity cells that have a predominantly phagocytic activity against pathogens and necrotic cells. They are also capable of regulating the adaptive immunity and inducing an inflammatory process [96,97].

The functional activity of macrophages in tissues is determined by the perception of various external signals due to the expression of many pattern recognition receptors, including Toll-like receptors (TLRs), and can lead to their further polarization [96,98].

The most currently discussed classification of macrophages is based on the identification of M1 and M2 phenotypes in cell populations, which are directly related to their pro- and anti-inflammatory properties, respectively [99]. According to this paradigm, macrophages can switch from a naive M0 state to a pro-inflammatory (M1) or anti-inflammatory (M2) phenotype, depending on the type of differentiation stimuli. In addition, it is possible to differentiate directly from M1 to M2 and vice versa [100].

The transition from the naive state of M0 to M1 is facilitated by the uptake of bacterial lipopolysaccharides (LPS), IFN-γ, and granulocyte-macrophage colony-stimulating factor (GM-CSF) molecules. The pro-inflammatory properties of M1 macrophages are explained by the fact that they secrete pro-inflammatory cytokines IL-1β, IL-12, IL-18, IL-23, and TNF-α. In addition, their participation in the immune response is associated with the expression of TLR-2, TLR-4, CD80, CD86, and MHCII molecules on the cell surface, which are essential for their interaction with other immune cells. M1 macrophages are also involved in the recognition and destruction of cancer cells [100].

Alternatively, activated M2 macrophages have potent anti-inflammatory properties. Tumor-associated macrophages (TAM) often exhibit the M2 phenotype. These cells are active phagocytes involved in extracellular matrix remodeling and angiogenesis. They suppress the immune response, thereby contributing to tumor progression [100,101]. Activation of macrophages towards the M2 phenotype can be caused by antigen–antibody complexes, invasive parasites, components of the complement system, apoptating cells, various interleukins (IL-4, IL-13, and IL-10), and transforming growth factor beta (TGF-β). Activation by these molecules stimulates macrophages to increase the secretion of IL-10 and decrease the secretion of IL-12, characteristic of the M2 phenotype [102].

The M1/M2 paradigm is often criticized as being too simplistic. However, this paradigm conveniently reflects the most phenotypically distant, i.e., polar states, of macrophage differentiation, and this terminology is often used in research descriptions [100].

To understand the topology of the antitumor effect associated with the migration of macrophages in the bloodstream, studies on the distribution of intravenously injected macrophages labeled with indium isotopes were carried out. It has been shown that radioactively labeled cells were concentrated in the lungs immediately after injection, after which they moved to the liver and spleen [103]. Finally, macrophages escaped from the lungs, liver, and spleen, but moved no further into the tumor [104].

Based on biodistribution results indicating the presence of a signal in the lungs and liver, researchers began to target cancers in these organs. In other studies, macrophages were injected directly into the tumor site, but in most trials, the effectiveness of the chosen therapy was low [104].

This may explain a significant number of clinical trials of macrophage therapy for the treatment of lung cancer, renal cell carcinoma, colorectal cancer, liver metastasis, etc. Currently, there are significantly low number of clinical trials dedicated to macrophage therapy of melanoma. In a recent review by Chatziioannou et al., the authors paid attention to melanoma treatment through the suppression or repolarization of TAMs using various molecules of agonists and antagonists of receptors, blockers, and activators of signaling pathways. The authors indicated that TAMs promote neoangiogenesis, lymphangiogenesis, and degradation of the extracellular matrix [105]. Despite the abundance of approaches used in the described clinical trials, there was no information on the use of isolated macrophages against melanoma. In this part of the article, we describe the experience of using ex vivo cultured macrophages in melanoma immunotherapy, and also pay attention to CAR modifications and the possibility of macrophage repolarization.

### 7.1. Cultivation of Cytotoxic Macrophages Ex Vivo

Hennemann et al. [106] evaluated the ability of cytotoxic macrophages to induce an antitumor response in patients with malignant melanoma in a clinical trial published in 1997. This was one of the first reported trials of macrophage therapy for this type of cancer. The therapy was based on ex vivo differentiation of human peripheral blood monocytes under the influence of GM-CSF and IFN-γ into cytotoxic macrophages with their subsequent administration to the patient in combination with different adjuvants. It was shown that side effects of the cell transfer therapy were mild, and no dose-limiting toxicity was observed. The results of the study allow for judging the potency of the proposed method for generating cytotoxic macrophages in vitro from monocytic precursors mobilized with GM-CSF. However, according to the results of the study, despite the large numbers of successfully generated and activated macrophages, no significant antitumor effect that could lead to the remission was observed. In the study group of 12 people, there was no significant clinical response, SD was observed in 8% of participants within 6 weeks, but despite this, the tumor continued to progress after the end of immunotherapy. NAE was observed in 100% of patients, predominantly in a mild form (Grade I–II toxicity). The authors suggested that the possible inefficiency is related to the choice of IFN-γ as an inducer of the activation of cytotoxic macrophages.

Additionally, a follow-up study was conducted by Hennemann et al. [107] in 1997. They initiated a new phase I trial to evaluate the impact of adoptive immunotherapy with IFN-γ and lipopolysaccharide (LPS)-activated macrophages on patients with various solid tumors. The lipopolysaccharide proposed by them in combination with IFN-γ is a bacterial endotoxin that contributes to the activation of macrophages due to the induction of TNF-α despite the evidence of a possible synergistic effect of IFN-γ and LPS [108]. Only one of the nine patients was diagnosed with malignant melanoma. The study also included patients with colorectal carcinoma, renal cell carcinoma, pancreatic cancer, and lung cancer. In many ways, this study repeated the results of the previous one where the effective differentiation of monocytic progenitors into macrophages under the influence of GM-CSF (up to 15 × 10^8^ cells as a single dose for a patient) and a mild form of toxicity (up to the established limit of a single infusion) were shown. However, the obtained results also indicated the ineffectiveness of the treatment of melanoma and other previously mentioned types of cancer, since no clinical responses were observed in the patients. SD at 12 weeks was observed in only 4.5% of participants with colorectal cancer, with NAE varying in the number of manifestations and in the severity of side effects, depending on the dose of injected cells. In a mild form (Grade I–II toxicity), NAE manifested in no more than 45% of the patients. Grade IV toxicity was shown only in 4.5% of the patients.

### 7.2. CAR-M Therapy

The success of CAR-T therapy has led many investigators to use CAR in combination with other cell types. CAR-expressing cells represent a major class of cellular immunotherapy that programs immune cells to recognize tumor-associated antigens and initiates specific antitumor response [109]. Macrophages, which, unlike T-cells, can enter the tumor microenvironment, are promising candidates for CAR modification. Such CAR-macrophages (CAR-M) are capable of initiating antigen-specific phagocytosis, secretion of cytokines, and destruction of cells expressing target antigens [110].

Unfortunately, there are currently no clinical trials of CAR-M for the treatment of melanoma. However, phase I clinical trials are ongoing in a variety of other tumor models [NCT04660929]. This is the first human study of adenovirus-transduced autologous macrophages engineered to express a chimeric anti-HER2 antigen receptor to target solid tumors overexpressing the HER2 antigen. A clinical trial has also been launched to determine the antitumor activity of CAR macrophages in patients with breast cancer [NCT05007379].

### 7.3. M2-to-M1 Repolarization

Despite the limited range of clinical trials associated with macrophage repolarization therapies, there are a number of promising studies whose proposed approaches could form the basis of a new wave of clinical trials.

Such studies, including the work of Chen et al., investigating chloroquine (CQ), which promotes the repolarization of TAMs, have shown that CQ, originally an antimalarial drug, can function as an antitumor immunomodulator that switches TAMs from the M2 phenotype to the M1 anti-inflammatory phenotype. M2 macrophages have a pronounced phagocytic activity, while M1 macrophages are mainly associated with inflammation and have a reduced phagocytic ability. Thus, it is reasonable to suggest that CQ promotes polarization of M1 macrophages by inhibiting phagocytosis via the pH increase in lysosomes. Although lysosomes have been regarded as the cell waste disposal system, a growing body of evidence indicates that lysosomes perform a much wider range of functions, including intracellular secretion, signaling, and energy metabolism [111]. Notably, lysosomal pH regulates both waste disposal and other lysosome functions. Thus, CQ treatment can alter the non-destructive functions of lysosomes by changing the pH, thereby restoring macrophage phenotype and function [112].

Macrophages as the basis of cellular therapy have indeed gained the opportunity to become the trend of a new wave of laboratory and clinical research. The first approaches to culturing cytotoxic macrophages in vitro had limited efficacy. With a single administration of the maximum allowable dose of cells to a patient, the therapy didn’t have a significant antitumor effect. However, despite the experience of inefficient use of macrophages in the past, with the help of modern cell engineering methods, in particular, CAR technology, macrophages can be used to implement a targeted antitumor response due to the natural ability of macrophages to invade the tumor and CAR-mediated recognition of tumor-associated antigens. Repolarization of macrophages in the tumor microenvironment may also be promising, but this approach requires more targeted studies and clinical trials.

The results of clinical trials, including the use of cell-based therapy and combined therapy for melanoma, are presented in Table 1.

## 8. Conclusions

The methods of melanoma cell therapy discussed in this review require further study and resolution of many problems, such as minimizing serious side effects, as well as optimizing methods for obtaining cell populations. In addition, one of the main problems of cell therapy is safety, for example, in relation to CAR-T therapy. However, some drugs have already received FDA approval for further clinical trials. Based on the data obtained from clinical trials in the field of melanoma, which are shown in Table 1, TIL therapy is the most demanded. The most promising new direction is the extracellular vesicles isolated from dendritic cells. This claim is supported by successful in vivo studies as well as clinical trials using Dex as cancer vaccines in comparison with DC-based vaccines. Dex function as immunomodulators and initiators of enhanced antigen-specific immune response of T-cells and NK cells. However, it is important to note that these therapy approaches are adjuvant and can be prescribed only after the main stage of cancer treatment.

## Figures and Tables

**Figure 1 ijms-24-02413-f001:**
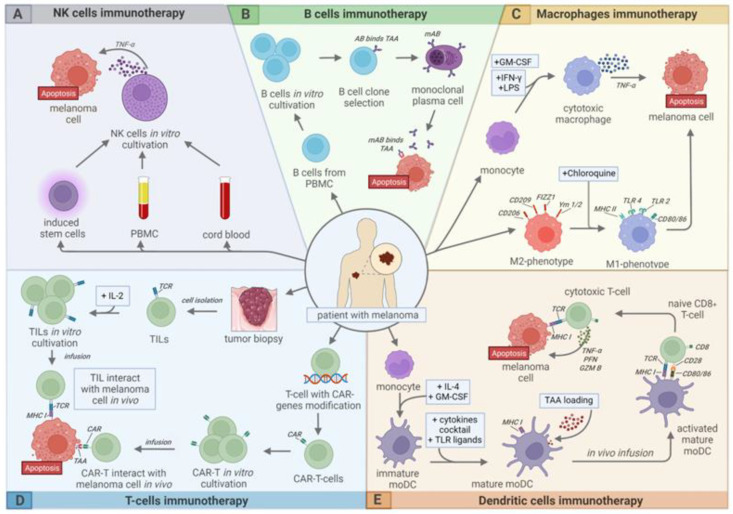
Modern approaches to cell-based immunotherapy of melanoma. (**A**)—NK cell therapy, (**B**)—B cell therapy, (**C**)—Macrophage therapy, (**D**)—T-cell therapy, (**E**)—Dendritic cell (DC) therapy. This picture is licensed under a Creative Commons Attribution 4.0 International License.

**Table 1 ijms-24-02413-t001:** Clinical results of single or combined cell-based in the treatment of melanoma.

Therapy Type	Drug	Participants	Efficiency, %	References
TIL therapy entry 2	Prior IL-2 Cyclophosphamide IL-2 Fludarabine 1200 total body irradiation	26 (18-have not received prior IL-2)	CR-38.88 NAE-100	NCT00314106
		(8-received prior IL-2)	CR-37.5 NAE-100	
	Aldesleukin CD8+ enriched Young TIL	12 (7, Aldesleukin)	CR-0; PR-14.28 PD-85.71; SD-0 NAE-100	NCT01118091
		12 (5, Adoptive Cell Therapy)	CR-0; PR-20 PD-80; SD-0 NAE-100	
	Prior IL-2 200cGy of total body irradiation TIL IL-2	23	CR-4.34 PR-39.13 NAE-100 ACM-4.34	NCT00096382
	No prior IL-2 200cGy of total body irradiation TIL IL-2	3	CR-33.33 PR-66.66 NAE-100	
TCR therapy	Tebentafusp (IMC10gp)	252	NAE-100 ACM-34,285	NCT03070392
	Decarbazine Ipilimumab Pembrolizumab	110	NAE-99.09 ACM-51.35	
CAR-T therapy	Anti-VEGFR2 CAR CD8 plus PBL Cyclophosphamide Aldesleukin Fludarabine	24	Terminated (No objective responses were observed.) NAE-100	NCT01218867
NK therapy	FT500 Nivolumab Pembrolizumab Atezolizumab Cyclophosphamide Fludarabine IL-2	37	ORR-24 ACM-<3	NCT03841110
	Autologous NK-cells, Lymphodepleting chemotherapy	8	No clinical response	[76]
B cell therapy	Hybrid cell vaccination Chemotherapy Immunotherapy IFN-α and/or IL-2Irradiation	16	CR-6; PR-6; SD-31; NAE-88	[69]
DC therapy	AdVTMM2/DC Vaccination IFN	35	ORR-29 CR-12 NAE-51 NED-36.36	NCT01622933
	Recombinant CD40-ligand Therapeutic autologous dendritic cells	62	OS-19	NCT00053391
	Exosomes from DCs DEX-based vaccine	15	ORR-50	PMC2657211
	Tumor lysate, particle-loaded, DC Placebo	184	OS (63-experimental group; 35-placebo group)	NCT02301611
Macrophage therapy	IFN-γ activated Monocyte-derived tumor-cytotoxic macrophages infusion Recombinant human GM-CSF	12	NAE I–II-100 SD-8.3	[95]

OS—Overall Survival, ORR—Objective Response Rate, CR—Complete Response, PR—Partial Response, NAE—Number of Adverse Events, PD—Progressive Disease, SD—Stable Disease, NED—No Evidence of Disease, ACM—All Causes Mortality.

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
