# Peer review of "Cell Immunotherapy against Melanoma: Clinical Trials Review"

_ijms, 2023, doi:10.3390/ijms24032413_

Round 1
Reviewer 1 Report
Cell immunotherapy against melanoma: clinical trials review
Authors Filin I. Y. et.al
The manuscript summarized the literature on ongoing clinical trials of cell-based therapy and adjuvant therapy on melanoma. Overall, the manuscript is well written. A couple of points need to be added before consideration for publication.
Line 58: please explain in more detail “genetic predisposition” or review the literature.
LN-44 [NCT02360579] also has been submitted to the FDA. Please include the review.
Line 519: What is the abbreviation of NAE?
Reviewer 2 Report
Filin et al highlighted all the available adoptive cell-based immunotherapy for melanoma. Overall, the review is very well performed, which included all the current adoptive cell therapies against melanoma, but there are minor points the author would need to address.
Minor points:
1. It will be great if the author can include or discuss the possibility of using the stem-cell-like or progenitor T cells as one type of therapy for melanoma treatment.
